# Birth in shelters: Midwives' lived experiences in providing childbirth care amidst war in Gaza

**Sahar Hassan**[1]*, **Suha Baloushah**[2], **Berit Mortensen**[3]

**1** Department of Nursing and Master's Program in Women's Health, Faculty of Pharmacy, Nursing and Health Professions, Birzeit University, Birzeit, Palestine, **2** Department of Medical Science, Nursing Department, Al-Aqsa University, Gaza Strip, Palestine, **3** Faculty of Health Sciences, Oslo Metropolitan University, Oslo, Norway

* sjamal@birzeit.edu

## Abstract

### Background

The warfare against the Gaza Strip in the occupied Palestinian territory, starting in October 2023, caused massive destruction of infrastructure, including homes and hospitals. Within two years (October 2023 to October 2025), most of the 2.1 million inhabitants were displaced, and more than 69,500 people were directly killed, of whom more than 70% were women and children. To enhance preparedness in maternal care during war, it is important to understand how midwives respond to emergencies in conflict zones. This study seeks to explore the experiences of midwives in delivering childbirth care to women in Gaza within shelters, tents, and amidst the rubble during wartime.

### Materials and methods

We employed a descriptive qualitative phenomenological design. We used a purposeful snowball sampling approach to identify midwives who had assisted women in giving birth in shelters or tents during the war. Face-to-face in-depth interviews were conducted with nine midwives between January and April 2025, and underwent reflexive thematic analysis.

### Results

An overarching theme was identified: "Midwives torn between fear and professional compassion while struggling for humanity in a ruthless war". Three additional themes were developed: "Unprotected in ruthless warfare", "Professional role and contradictive emotions", and "Challenges and potentials for midwives during war and emergencies".

**Data availability statement:** All relevant data are within the manuscript and its Supporting information files.

**Funding:** This study was funded by NORAD through NORHED II project (Midwifery Research and Education development in Ghana and Palestine (MIDRED) Ref. 70320) at Birzeit University. Authors: SH (from Birzeit University in Palestine) and BM (from OsloMet University in Norway) received the grant from NORAD. https://www.norad.no/en/. The funders had no role in study design, data collection and analysis, decision to publish, or preparation of the manuscript.

**Competing interests:** The authors have declared that no competing interests exist.

## Conclusions

The study recommends the development of clear preparedness policies aligned with midwives' scope of practice that support their professional role when caring for pregnant women in war, emergency, and humanitarian contexts. Given the volatile context of the oPt, Palestinian midwives should be actively involved in preparedness strategies for emergency response. They need guidance, support, necessary supplies, connection to backup resources, and training to assist births safely outside hospital settings.

## Introduction

Midwives have a critical role in providing sexual, reproductive, maternal, newborn, child, and adolescent health services in humanitarian settings and are included in the Minimum Initial Service Package for reproductive preparedness in crisis [1,2]. Nove et al. concluded from their analysis that midwives have the potential to substantially reduce maternal and neonatal mortality, as well as stillbirth, in Low-Middle-Income Countries (LMICs) [3]. The International Confederation of Midwives (ICM) has called on governments to include midwives in emergency preparedness programs and to enable them to provide care for women during emergencies and humanitarian crises [4].

In October 2023, the still ongoing warfare started against the Gaza Strip of the occupied Palestinian territory (oPt). A war that has caused massive destruction of infrastructure in all sectors, directly killed more than 69, 500 people, of whom more than 70% are women and children, with an estimated several hundred thousand indirectly killed from wounds, starvation, and lack of medical treatment [5–7]. The war has resulted in damage to approximately 81% of the road network, to all hospitals, more than 63% universities, 89% of WASH sector assets, while 92% of houses have been destroyed [8] and about two million were internally displaced multiple times across Gaza [9].

In 2022, 57.445 women gave birth in the Gaza Strip, where the fertility rate was 3.9 [5,10]. The latest UNFPA report indicated that approximately 17,000 births were recorded during the first six months of 2025, representing a 41% decline compared to 2022 [9]. The report also highlighted an increase in miscarriages, premature births, low birth weight, and neonatal mortality [9]. One in three pregnancies was classified as high risk. The violation of International Humanitarian Law (IHL) mounted in Gaza, where 36 hospitals were attacked and suffered severe damage to essential monitoring equipment and supplies, including diagnostic tools, basic lab analyzers, fetal monitors, resuscitation equipment, and incubators [9,11]. As a result, many women faced significant barriers in accessing safe childbirth care within hospitals and were often forced to give birth in unsafe and unhygienic environments in shelters among the rubble [11]. Furthermore, fear, anxiety, depression, and the death of a loved one are all causes of a wide range of mental health problems among pregnant women [11].

Gaza is a narrow strip, 365 square Kilometers wide and 41 kilometers long by the Mediterranean Sea. It is the home for about 2.2 million Palestinians, the majority of whom are refugees [12]. Gaza has been under complete lockdown since 2006 and has experienced four wars since 2008 (2008–2009, 2012, 2014, and 2021), which resulted in massive destruction of infrastructure in all sectors [13].

Studies from previous wars have documented the experiences of Gaza women struggling to access hospitals to give birth during bombardments and airstrikes, as well as on the unpreparedness of midwives who were approached by laboring women in their homes to assist them in giving birth safely [14]. A nine-year surveillance study revealed a steady increase in low birth weight, birth defects, and preterm births, as well as the potential reproductive morbidities among Gaza women due to the accumulation of high levels of weapon-derived metals in bodies [15]. A recent study conducted among 500 pregnant women in Gaza reported an association between war-related stressors and adverse maternal and neonatal outcomes during 2023–2024, with a higher prevalence of low birth weight, increased rates of maternal anemia due to food insecurity, and reduced accessibility to antenatal care [16].

The IHL are meant to protect civilians and health care providers during war. But recent wars are increasingly targeting and affecting healthcare providers (HCP), working in conflict zones [17]. Many were attacked while working in hospitals, lacked proper training or support, endured physical dangers, threats to their families' safety, and experienced financial hardship. These circumstances forced many to migrate after being targeted with violence and witnessing horrors of conflict everywhere [18,19]. To improve maternal care preparedness in a world increasingly facing conflicts and violations of IHL, the lessons learned from midwives in Gaza can provide important knowledge. This study is crucial as it generates context-specific evidence on delivering childbirth care in extreme and resource-constrained conflict settings, where formal health systems are disrupted. It seeks to explore the experiences of midwives in delivering childbirth care to women in Gaza within shelters, tents, and amidst the rubble during wartime.

## Materials and methods

### Study design

To describe and understand midwives' lived experiences of assisting pregnant women during childbirth in shelters during war, we employed a descriptive qualitative phenomenological design. This approach seeks to explore phenomena as they are experienced by individuals in their everyday realities [20], aiming to capture the deeper meanings embedded in Palestinian midwives' continuous care for women during childbirth amidst bombardment and severe wartime conditions [21,22].

### Study setting

The war in Gaza broke out in October 2023. The study was conducted during the following twenty-two months across different areas of Gaza, including south, middle, and north, where displaced midwives could be accessed. We focused on midwives working among the displaced population and assisted pregnant women to give birth in shelters and tents during the war, regardless of their geographical location.

### Participants and data collection

A purposeful snowball sampling approach was employed to identify midwives who had assisted women in giving birth in shelters or tents during the war. Recruitment was based on the criteria that participants had assisted at least one pregnant woman in giving birth in a shelter or tent and were willing to share their experiences. To ensure maximum variation, midwives of different ages, educational backgrounds, and geographical areas were included.

Face-to-face in-depth interviews were conducted between January and April 2025 by a midwife researcher (SB) based in Gaza. All interviews were carried out in Arabic, semi-structured by an interview guide that was developed and reviewed by all authors. Interviews were audio-recorded and transcribed verbatim (in Arabic) immediately after each interview. The

interviews were conducted in locations that were safe and convenient for both the midwives and the investigator, ensuring privacy for sharing experiences. Each interview lasted between 30 and 60 minutes. No personally identifiable information was recorded or stored alongside the transcribed data. Oral consent was obtained prior to each interview, and data collection continued until variation and richness of information were obtained. Each interview began with a broad, open-ended question, such as *"Could you tell me about your experience providing care for women giving birth in a shelter or tent during wartime?* Probing questions were used to clarify midwives' descriptions, with prompts such as: *"What did you mean by that?", "Could you explain more?", "Can you give me an example?"* The final data set consisted of nine transcripts of recorded interviews (S3 Appendix: Interview guide).

To ensure relevance, and content quality, the interviews were carefully read and subsequently discussed by the first and second researchers (SH and SB), both are experienced Palestinian midwives with extensive backgrounds in providing care in politically unstable humanitarian settings. Credibility and dependability were further enhanced through regular discussions and guidance from the first author, an academic nurse-midwife and researcher with extensive experience in midwifery. This process involved debriefing, maintaining an audit trail throughout the data collection, and supervising the transcription of the interview to ensure confirmability. The comprehensiveness and clarity of responses, as well as the need for further elaboration on certain points, were identified and discussed among the first two researchers to inform subsequent interviews.

To ensure the credibility of the study, all interviews were conducted face-to-face to establish a close and trusting relationship with midwives. Despite the ongoing war, sufficient time was spent to collect the data despite the ongoing war situation. To enhance validity, there was dedication to data collection to ensure variety, depth and richness of the narratives. Interviews were independently coded and analyzed by three qualitative researchers, with highly consistent results across all analysis.

## Data analysis

Data were analyzed inductively, and independently by the three researchers using the six phases of reflexive thematic analysis as described by Braun and Clarke [23] 1) familiarization of data 2) coding 3) generating initial themes 4) reviewing themes, 5) refining and naming themes and 6) writing up analysis, with a focus on capturing the lived experiences of midwives [20]. The interviews were transcribed verbatim by a research assistant familiar with qualitative research methods, under the close supervision of the first author, who cross-checked the transcripts against the audio files to ensure completeness. This was followed up by the second author, who carefully reviewed the full transcript of each interview to verify accuracy and confirm the content. The first and second authors conducted their analysis using the original Arabic transcripts, while the third author worked with a version translated to English by the Artificial Intelligence software, Perplexity [24]. The two Arabic-speaking authors reviewed and approved the translated transcripts for accuracy and contextual meaning before the third author initiated the analysis. The content of the English transcripts was compared to the Arabic and confirmed by the co-authors to enable the last author to conduct the analysis. For coding and analysis, the first author used Tanguette software [25], while the second author used MAXQDA software [26] and the last author used Excel. The two Arabic-speaking authors (SH and SB) became familiar with the original data by repeatedly reading and discussing content for each interview, typically every one to two interviews, during the data collection process, while the last author got familiarized with the translated transcripts. The three authors (who are all experienced qualitative researchers and PhD holders with a midwifery background and familiar with the Palestinian context) coded the transcripts separately, revised and collated codes to identify patterns and similarities across all interviews, organized meaningful units, developed initial sub-themes, and subsequently refined, defined, and named the sub-themes. This was followed by a thorough review and reflexive discussion among the three authors, during which high consistency was achieved. Codes, subthemes, and themes were re-examined, organized, and refined through this collaborative process (See S1 and S2 Appendix). We adopted a phenomenological research approach, where we applied a thematic approach due to its

flexibility in identifying repeated patterns of meaning within qualitative data, an approach commonly used in nursing and midwifery research. Thematic analysis grounded in descriptive phenomenology is particularly useful when exploring lived experiences, as in this study [20]. Understanding the essence of the phenomenon was achieved by engaging deeply with the data to identify meaningful patterns, focusing on the richness and complexity of the narratives rather than counting frequencies of specific responses [20,21].

## Ethical consideration

The study was conducted in accordance with the Declaration of Helsinki [27]. A written consent statement was included on the first page of the interview guide and was read by all participants, with approval provided by each midwife. At the start of recording, the interviewer (second author) read the consent aloud, and each midwife gave recorded oral approval before the interview commenced. Oral informed consent was obtained from all participating midwives, who were assured confidentiality and their right to withdraw at any time. No names or identifiable information, such as IDs or phone numbers, were included in the audio recordings or transcripts. Oral consent was clearly stated (in Arabic) in accordance with the written form and was audio-recorded recorded at the beginning of each interview. All audio recordings were deleted after the interviews were fully transcribed. Each transcript was assigned a serial number to ensure anonymity. Ethical approval for the study was obtained from the Ethical Research Committee at the Faculty of Pharmacy, Nursing and Health Professions at Birzeit University, West Bank (BZUPNH2401).

## Results

### Characteristics of midwives

The study sample comprised nine midwives from the northern, middle, and southern Gaza Strip, all female. Their ages ranged from 29 to 50 years. Five midwives were married and had children. All midwives held a midwifery degree, either a Bachelor's or a Higher Diploma, and had between 5 and 26 years of midwifery experience. The midwives' experience varied widely (mean ± SD: 12.67 ± 7.48 years), contributing to the richness of the data. Two midwives also held a Master's degree. All had been displaced multiple times from their original homes and were currently providing care to pregnant women, children, and adults in shelters, tents, or temporary medical points (Table 1).

### Themes and subthemes

Our findings are presented as one overarching theme, three main themes and ten subthemes (Fig 1). The overarching theme "Midwives torn between fear and professional compassion while struggling for humanity in a ruthless war"

**Table 1. Characteristics of midwives who participated in the study.**

| Midwife | Education | Years of Experience (Years) |
|---|---|---|
| Midwife 1 | Bachelor Midwifery, Master health management | 9 |
| Midwife 2 | Diploma Midwifery, Bachelor health management | 5 |
| Midwife 3 | Bachelor nursing, high diploma midwifery | 17 |
| Midwife 4 | Bachelor midwifery | 19 |
| Midwife 5 | Bachelor midwifery, Masters public health | 18 |
| Midwife 6 | Bachelor midwifery | 6 |
| Midwife 7 | Bachelor midwifery | 7 |
| Midwife 8 | Bachelor midwifery | 7 |
| Midwife 9 | Bachelor midwifery | 26 |

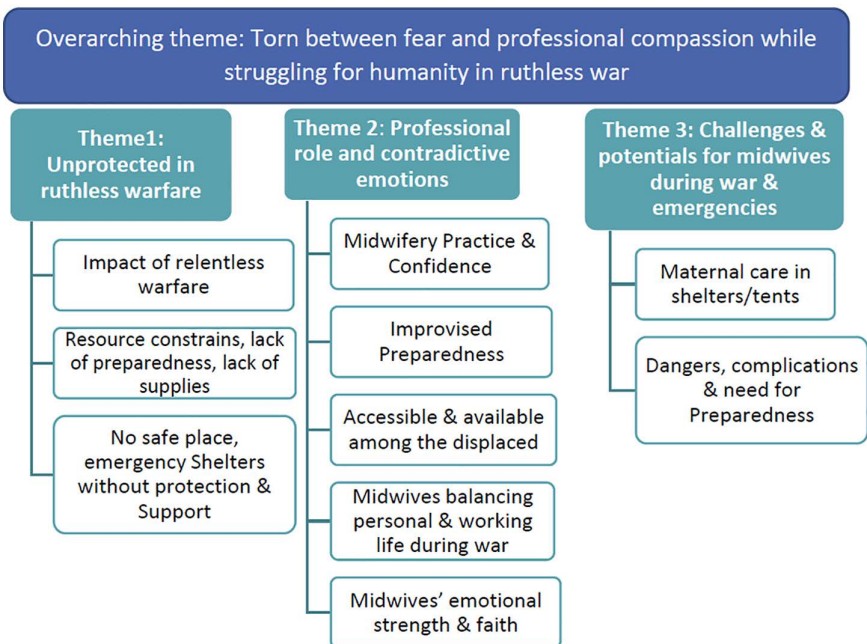

**Fig 1. Themes and sub-themes.**

describes the emotional and professional contradictory experiences reflecting these midwives' stories of life-threatening challenges and lifesaving achievements. The midwives' urge to help women bring new life with dignity and humanity in an alien environment, unprotected by humanitarian law. As one of the midwives describes:

*"I was called to help a pregnant woman in our neighborhood. At first, I was scared because the quadcopter was flying behind me at that moment, and I was afraid something would happen. I was afraid of the situation, but for me it didn't matter whether I lived or died because I had just survived. I was afraid of hearing people screaming and seeing my house being destroyed. Shall I run to help injured people on my way, or shall I continue to assist the woman to give birth. I felt like I was caught between two fires"...... "My experience was mixed with emotions, but I'm very happy I was able to save a life. When you see the baby crying, you feel you've accomplished something great".* (MW2)

**Theme 1: Unprotected in ruthless warfare.** Midwives recalled the *impact of warfare* on their working environment, where they were obliged to assist women giving birth in shelters amid ongoing bombardments, fear, and a pervasive sense of insecurity. They described how they felt unprotected under bombardments when no ambulances were in reach, and shelters were attacked.

*"I examined her and she was fully dilated..., and of course, it was very difficult for us to move; anyone who moved was targeted. Calling an ambulance was very difficult; even if the ambulance knew about the birth, they wouldn't know how to get there. Of course, I had to deliver her at the school (shelter), and I did .... the Israelis reached the school by the afternoon, so we all had to flee".* (MW5)

Midwives described the profound *impact of warfare* on daily life, particularly for women, which was marked by stress, fear, and ongoing challenges. They reported several difficulties experienced by pregnant women, including neglect of their

health and follow-up care, which women often perceived as less important than the threat of bombardments surrounding them. Midwives also highlighted the economic consequences of war, which restricted access to essential food, vitamins, and family planning methods. Additionally, they observed an increase in early marriage among adolescent girls.

*"Frankly, most pregnant women are neglected. For example, I know she is anemic, but when I ask her if she has taken iron supplements, she says no. I don't have the money to buy iron supplements, so we don't buy them".* (MW1)

Midwifery care provided by Gazan midwives closely resembled the model of midwifery-led care. They provided holistic support, during pregnancy at medical points in shelters, were available in emergency births or accompanied women to the hospital when possible, and documented birth details for later birth certificate registration. Midwives reflected on leading childbirth outside hospital settings, often with *resource constraints, lack of preparedness, and lack of supplies*. All midwives reported *being unprepared and lacking essential equipment* while assisting women during childbirth. They recalled having no delivery kits, antiseptic solutions, emergency medication, or even gloves, as well as no transportation, and often no signal to call the ambulance. Many midwives recalled assisting births in overcrowded tents where all family members were present, as it was not safe to ask them to go outside. In such situations, midwives improvised by creating some privacy with blankets, often with the help of family members. They described the profound stress and feelings of shame experienced by pregnant women who had to give birth under these extremely difficult circumstances. A midwife reported assisting a woman who went into labor in the middle of an overcrowded street, in a mud-soiled area without sanitation or essential equipment, where the baby was already about to be delivered. The midwife was compelled to provide care and assist with the birth under these unsafe and chaotic conditions.

*"I heard women shouting in the street. I ran out and said, "Maybe she's giving birth." All I had were my hands and a scalpel, and I found the woman fully dilated under the blanket in the street. I couldn't move her... I took another blanket that was used to sell vegetables by the near point. The head was completely out, so I pulled the baby out. It was a girl. I had a thread and a scalpel with me, so I tied the thread and cut the umbilical cord about 6-7 cm away from the navel. Because I wanted to move her to another place, I didn't have any sterile tools, so I cut with the scalpel, wrapped the baby in the blanket, took her, and put her in a "kara" (a cart pulled by a donkey). We asked him to take her to the nearest medical point by the UN".* (MW9)

*"I assisted a woman to give birth in a tent where the ground was all muddy and polluted, with the sound of warplanes overhead".* (MW2)

Despite the absence of *protection in emergency shelters* and the absence of any truly safe space or place, midwives strived to create a dignified birthplace whenever they were called to assist, whether on rubble, in schools, shelters, or tents with muddy floors, often surrounded by shrapnel. Midwives emphasized that no place felt safe.

*"People knew we were in that school and that we were midwives. Around 4:00 a.m.,.... came to the classroom and told us there was a birth case downstairs. I went down, but the school I was staying in didn't have privacy, so I tried to move from school to school until I found... it was very difficult. There were shells and shrapnel. We made openings in the walls to move between schools until we reached the medical point in the neighboring school, which had some privacy. I examined her; she was 8 cm dilated, the cervix was soft, membranes intact……..the woman's mother was with her".* (MW6)

**Theme 2: Professional role and contradictive emotions.** Midwives reported their experiences of being called to assist women during childbirth in extraordinarily unsafe situations with a sense of professional pride, self-confidence,

professional duty, and unwavering dedication to women. Their stories reflected remarkable courage in *conducting midwifery practice with confidence,* responding to women's needs despite the dangers.

"*The schools were evacuated, everybody left, and people said the school should be evacuated. I was assisting a woman to give birth. I told them I would stay with her, and they all left. I stayed with the woman and the baby, and I took out the placenta, and I left the school with the woman, just delivered, and her baby. I told her, "Come with me, let's go." The woman asked me to leave and keep her and the baby there. I told her, "No, you and I are leaving together. Just after we left, they hit the school…"* (MW8)

Midwives reflected on their wartime practice, noting that their difficult experiences strengthened their confidence and sense of pride in their professional role. They described how providing care under bombardment affirmed their dedication and commitment to women, often as the only available healthcare providers. Working in such challenging circumstances also enhanced their utilization of midwifery skills by providing non-medical interventions to prevent complications. As these midwives described:

"*The delivery kit had no drugs. I relied on non-medical interventions like frequent bathroom breaks, massages, and close observation until the woman was out of danger.*" (MW5)

"*The baby was fine, crying. I dried the mouth and nose, wrapped the baby in a clean towel, and did skin-to-skin contact to stimulate the uterus to contract since there was no medication..... skin-to-skin helped with placental separation. I clamped the cord and started breastfeeding, which helped prevent postpartum hemorrhage*" *(*MW7)

These circumstances encouraged them to rely more on natural, non-intervention practices and to avoid unnecessary procedures, such as rupturing the amniotic membrane, performing episiotomies, or perineal tears, to safeguard both mothers and the newborns.

"*We attempted to employ natural practices such as massage every quarter of an hour and close observation. I kept my eyes on her and did not leave her side for any reason.*" *(*W5)

"*The baby was placed on the mother's stomach until the placenta came out, without clamping. The baby took warmth from the mother and benefited greatly from skin-to-skin contact.*" (MW8)

Another midwife recounted keeping a woman in her own tent for four hours after delivery, assisting her with breastfeeding, providing herbal tea, monitoring uterine contractions, and ensuring the well-being of both mother and newborn. This was very deeply appreciated by women, who noted that the care they received from the midwife was far better than the care they experienced in hospitals during their previous births before the war.

"*While she was still in my tent, I helped her to breastfeed and prepared a warm herbal drink for her, such as fenugreek and similar remedies.*" (MW4)

In the absence of any policy guiding midwives in this humanitarian emergency and chaotic context, midwives assumed full accountability and responsibility for the births they assisted in tents. They carefully documented key details of each birth. Some midwives also reported accompanying women to the nearest medical point or hospital to complete their duties, such as ensuring placental delivery, as they were uncertain whether hospitals have sufficient staff to continue care.

"*I went with her to the hospital, stayed with her, and delivered the placenta at the hospital. I stayed with her, gave her all the care she needed, and left the hospital with her in the morning".* (MW5)

*"I wrote down the mother's name, her ID number, her age, the baby's gender, the time of birth, the estimated weight, all the details, the place of birth, who was with us as witnesses at the birth, and I gave the paper to the mother and wrote my name on it and told her to take it to the birth registration office or, after the war, to hand it over to them so that the baby could be registered".* (MW8)

All midwives had to *improvise preparedness,* as they clearly reported that they were unprepared and lacked essential equipment. They requested delivery kits, and some noted inadequate training for handling childbirth outside hospital settings. Despite these challenges, they identified their priorities during childbirth, and demonstrated creativity in using whatever resources were available to facilitate safe and dignified deliveries. They emphasized the need for regular training and consistent access to sufficient delivery kits, particularly given that they work in a humanitarian setting where warfare has been frequent over the past decades. During the war, the midwives were provided with simple kits and equipment.

The lack of essential equipment and the severe shortage of resources forced midwives to improvise by using household and kitchen utensils during childbirth. These included kitchen knives, ordinary sewing threads used for knitting tents, flame from fire or salt and water to sterilize scissors, and, at times, examining women without gloves.

*"We didn't have a delivery kit at the time. Supplies were scarce, and we had to improvise with what was available. After this incident, there was more focus on ensuring every medical point had at least four delivery kits"* (MW1)

*"Bring me a kettle. The woman was almost fully dilated... I put some water in it and added salt. I put the threads in it to sterilize them. I told them to bring me scissors, but they said they didn't have any. I took a knife and put it on the fire to sterilize it".* (MW2)

*"I boiled a scissor and sterilized it with water, then made a small perineal cut."* (MW7)

Midwives were always *accessible and available* to displaced pregnant women and their families. They were called by family members to assist a woman in labor, or women themselves came to the midwife's tent seeking help. In one case, a midwife reported being called at night by the nurses from the nearest medical point to assess a woman in labor. At times, relatives or neighbors informed families about the presence of midwives in the shelter or directed pregnant women to the midwife's tent in the same shelter. One midwife described assessing ten women to determine whether they were in labor and providing post-cesarean care.

*"I took care of almost two births as normal deliveries; from the moment the mother went into labor until the baby was born, and I took care of the baby. But there were other cases, maybe 10 cases, that needed to be assessed. I also provided postnatal care in cesarean cases,…"* (MW5)

All midwives described how they strived to *balance professional and personal demands* as they were all on the run for shelter from warfare. They reported being displaced multiple times after their homes were destroyed, and each time they tried to arrange a new tent or shelter for their families, they were displaced again, losing the few basic items they had, such as blankets, tents, clothes, and cans of food. All midwives described, with sorrow and tears, the loss of family members who were killed either before evacuation or during displacement, as well as the difficulty of obtaining food for their families. *"We were displaced in five or six shelters". (MW6).*

*"I have no clothes. I'm fleeing from war, which means I have no home and no shelter. Everywhere I go, I leave my clothes and belongings behind and run away..." (MW8)*

A few midwives reported having family members who were disabled or severely injured and for whom they were responsible. *"My son and husband have injuries from the war". (MW3).*

Despite the profoundly unsafe circumstances, midwives continued to work and provide care for women around the clock, whether in destroyed hospitals, medical points or in their tents in displaced communities, often with partial and irregular salaries, and sometimes with none at all. They reported struggling to secure basic food items for themselves and their families due to extremely high prices. For instance, one midwife reported paying about 30 USD for just two cucumbers, while a kilogram of tomatoes cost 35 USD. Midwives reflected on their work and living conditions with pain, tears, sorrow, sometimes expressed in only a few words or even in silence. They described the dangers of crossing dangerous areas under ongoing bombardments or sudden evacuations to reach damaged hospitals, where they often worked for 24 hours or more.

*"Every day, I walk to work on a road that is bombarded with shelling. Transportation is difficult because it is unavailable and the distance is long. I walk for an hour and a quarter to get to work."* (MW3)

*When I was in Rafah, I worked 24 hours a day and delivered 100 babies. I never got tired or discouraged. I helped all the women, helping them so they wouldn't suffer. They would say, "Oh God, what wonderful service, what wonderful..." And in war..."* (MW4)

One midwife reflected on the challenge of leaving her small children alone, or with an injured husband in the tent, while she searched for food or fulfilled her professional duties. Midwives also witnessed poor conditions during childbirth, such as babies being delivered without protective clothing. They described the stress and sense of insecurity they experienced while trying to reach women in labor. One midwife described both midwives and laboring women as "victims", forced to work underprepared and with no or minimal resources, while women deserved to give birth under better conditions. Another midwife described her experience of assisting women in labor as "tough" and "harsh" for both the woman and the midwife. A few midwives expressed mixed feelings of pride at being present to help, hope for laboring women, yet deep awareness of harshness caused by poor conditions.

*"Felt like I was under stress. I wasn't thinking about whether I should go or not. In the end, you find yourself going even if there is a risk, because someone needs you. Your profession is humanitarian and trustworthy."* (MW8)

In addition to expressing positive feelings about fulfilling their professional responsibility during wartime, midwives' emotions were contradictory as they also reflected on the negative impacts of war on their emotional wellbeing. They described psychological distress, physical exhaustion, despair of personal losses, and the toll on their inner selves. Midwives further reflected on the loss of family members, homes relatives, friends, and workplaces during the war.

*"I lost my dearest people. My aunt's daughter was like a sister to me. Suddenly, overnight, I heard that she had been killed. I lost my uncle's children and my aunts' children too. They were also killed because of the war.* (MW4)

*"Our current situation, the problems and the war, the mental and physical exhaustion, the financial situation, all of this has affected my mental state. I feel exhausted. I want to stop living. I feel like I need to rest too".* (MW6)

*"We laugh at ourselves for being so adaptable, but we're not adapting at all".* (MW1)

*"I was torn between fear and wanting to help. As a midwife, I love my job and wanted to help by any means. At first, I was scared of possible complications, like bleeding or the baby needing special care, but I had to do my best".* (MW5)

The narratives of the midwives reflected strong self-motivation to assist laboring women despite the surrounding dangers. Some reported that assisting women made them feel better, as the harsh situation itself motivated them to help even with minimal or no resources, particularly when they witness women laboring alone without access to hospitals or other

healthcare providers. Furthermore, midwives explained that they could not leave hospitals because of the severe shortage of human resources needed to respond to the overwhelming demands of care.

*"Thank God, my morale was high. Helping people improved my mood and motivation". (*MW5)

*"….we were on emergency shifts, 24 hours at a time, sometimes covering for staff shortages, which affected my family, my children and husband needed me, and I needed their support. The long hours and lack of staff negatively affected my social life ….It affected my social relationships, led to conflicts and problems due to long working hours and absence from my family."* (MW6)

**Theme 3: Challenges and potentials for midwives during war and emergencies.** Midwives reported their care for newborns and women during and after childbirth in shelters and tents. These childbirth experiences were characterized by a crowded environment, the emotional distress of laboring women, a lack of essential equipment, and constant fear of complications, ongoing bombardments, and potential dangers. Despite all the challenges, the midwives emphasized that their professional commitment, confidence, and resilience contributed to successful outcomes. One midwife described how giving birth under bombardments, amid the screams of victims, immediately and negatively affected the progress of labor contractions.

*"While giving birth, they (laboring women) hear people shouting and the sound of shelling. You feel that the contractions have disappeared, even though in most cases the cervix was fully dilated. The woman say, "I don't feel the contractions." The contractions disappeared because of the stress you were under during childbirth".* (MW2)

Midwives also reported lack of access to ambulances and hospital, thus they had to provide care for women with complications, including those with previous cesarean section, preterm labor, or those carrying a precious baby after years of infertility.

*"I was at home, and my neighbor and friend went into labor after her house was bombed. She was seven months pregnant, and it was her first pregnancy after five years of infertility, so it was very difficult for her. The ambulance couldn't come. After the bombing, she was scared and in pain, but she thought it was mild. She was already 7 cm dilated. We called the ambulance several times, but they said they couldn't reach us because we were in a border area".* (MW8)

*"She's very stressed and afraid. She had married later in life and had waited several years before giving birth to her first child. The doctor had already scheduled her for a C-section for this pregnancy, so she was very afraid of this issue… The first thing I did was to calm her fears..."* (MW8)

*"I made a small incision in the perineal area because the woman had had a C-section for her previous birth and was worried for the baby. If it were up to me, I would have waited*". (MW7)

It is worth noting that all midwives were expected to attend regular working hours (or more due to emergency situation) at their workplace (if accessible), or nearby facility, or a local medical point, while simultaneously, providing emergency services around the clock in their own tents to displaced communities, where they had no choice but to assist women. All midwives emphasized the need for support at all levels, including professional guidance and preparedness, continuous training, proper provision of essential equipment, clear policies to protect them while delivering care in humanitarian settings, an effective referral system, recognition of their unique services, and regular, fair salaries.

*"There must be more support for midwives and pregnant women (during war)… Financial, social, psychological support, training and recreational courses for midwives".* (MW6)

## Discussion

This study explored the experience of Palestinian midwives assisting women who gave birth in tents and shelters during the prolonged Israeli warfare on Gaza that began in October 2023. The findings revealed how midwives were *Torn between fear and professional compassion while struggling for humanity in the context of war.* The midwives described being unprotected under ruthless warfare, risking their lives to save mothers and newborn. The Lancet series on Women's and Children's Health during Conflict, revealed how modern wars have increased disregard of International Humanitarian Law and highlighted risks for civilians, estimating a threefold increase in mortality among pregnant women living in areas of high intensity warfare [17,28]. Several UN-reports, Amnesty International and other researchers, have described the Israeli warfare in Gaza as a genocide, with women and children representing up to 70% of those killed and wounded [6,29–31].

The experiences shared by the midwives in Gaza adds important knowledge related to preparedness needs and the creative responses adopted by midwives during intense warfare. Licensed and educated midwives are essential for providing comprehensive, quality maternal and newborn care within health systems [32]. Pregnant women are facing increased risks under armed conflict and war; thus, midwives should be fully integrated into preparedness strategies [2,4,16,33]. The midwives in Gaza reported that they had to assist emergency births alone, outside any supportive system, as they were unable to reach ambulances or facilities. Such responses align with previous research revealing that, midwives within communities are often the first and only professional to reach out to women in labour when they are prevented from accessing hospitals during warfare [34]. The midwives in our study expressed a preference for being better prepared, equipped, and supervised, in line with findings from previous research on midwifery in fragile settings [1]. Although their requests for training and supplies were addressed during the war, but the kits they received were reported to contain insufficient emergency equipment and medications.

The midwives in our study described how they were called by people around them who knew they were professional midwives. They responded without hesitation to help in emergencies despite being unprepared and unequipped. Their professional role made them overcome fear, and they responded to emergencies by using their midwifery competencies and what was available at hand. Research from a previous war on Gaza in 2009 described how midwives could played a crucial role in the community, helping women in labour who could not reach hospital, nevertheless many were reluctant as they were not prepared and feared that they were restricted by law in providing care outside hospitals [14]. The midwives during the recent war did not mention fear of restrictions, which might imply a growing recognition of the profession's autonomy within the system. Considering the challenges experienced by the Palestinian midwives highlights the potential for developing preparedness strategies and policies more directly aligned with the midwife's scope of practice during war and emergencies. This is consistent with finding from the systematic review on midwife's roles in humanitarian settings conducted by Beek et al (2017), which revealed that midwives' activities and roles were lacking in the preparedness phase of the disaster management cycle.

The midwives described how they gave comprehensive midwifery care, from providing antenatal care in shelters, emergency labour care until the woman could be transferred safely to a health facility. Some focal medical points in shelters included essential medications and supplies for the displaced population, such as intravenous cannulas, fluids, commonly used analgesics, and antibiotics. Essential medications for obstetric care, such as Pitocin and magnesium sulfate, were not available in shelters or focal medical points. Despite the alien environment, midwives' care was in line with values and philosophy-qualities referred to the Quality Maternal and Newborn Care (QMNC)-framework [35] except for the absence of first-line essential drugs required for the management of obstetric emergencies. They relied on their core midwifery philosophy, optimizing physiological processes to prevent complications and avoid unnecessary medical interventions. The most urgent quality missing was sufficient for first line management of complications and referral possibilities. In the absence of emergency medications, they were often required to manage high-risk pregnancies and make serious professional decisions aimed at safeguarding and preventing further complication.

As in many other LMICs, Palestine has had a focus on improving midwifery models of care the last 20 years [36–38]. However, more attention should be given on preparing midwives in assist women in birth outside hospitals given that securing health facilities during war can no longer be taken for granted. Currently, modern midwives work in highly medicalized hospital environments where obstetricians are available, and unnecessary medical interventions are often implemented too early [39]. Creating home-like birthing environments, where midwives practice according to midwifery philosophy and values, would likely enhance their preparedness and ability to provide essential midwifery also during emergencies.

The UN Relief and Works Agency for Palestine Refugees in the Near East (UNRWA) provided a well-organized system for involving midwives in preparedness and response within the health system during crises [2], which continued to provide services for communities and support some midwives in focal medical points in shelters. Israel targeted UNRWA by sanctions and attacks, disrupting the delivery of lifesaving supplies to the refugees who depend on the organization [40]. Throughout the war, Palestinian Ministry of Health, the UN and several national and international health agencies worked to repair damaged ambulances and hospitals, establish field hospitals and makeshift clinics to serve the population, nevertheless these were repeatedly attacked and damaged causing an increasing number of women had to give birth outside a health facility [41].

## Strengths and limitations

This study has several strengths. One is the unique and rich information derived from midwives' experiences of providing emergency labour care alone in the context of war in Gaza. These insights could inform revisions of research-based guidelines and MISP by emphasizing preparedness and advocating for policies supporting emergency midwifery care outside facility settings. Another strength of this study is that the study was led and conducted by local researchers, who possessed a unique ability to gain trust and access eligible midwives. A limitation in this study is that all participating midwives shared stories that, in some way, ended well with a healthy mother and baby. Important information may have refrained from participating due to fear of being perceived as having failed. Additionally, midwives may have been reluctant to disclose cases with complications, as assisting women to deliver outside the hospital is considered an illegal practice. Consequently, they may have feared potential consequences or sanctions from the health system. It is therefore possible that midwives selectively shared narratives that ended without complications.

## Conclusion

The lived experiences of caring for pregnant women and assisting them to give birth in tents, shelters and amid rubble were vividly expressed by Palestinian midwives during the two yearlong warfare on the Gaza Strip starting in October 2023. The study highlighted the courage, rapid and creative response, and challenges faced by these midwives. They were torn between fear and professional compassion while striving to uphold humanity amid war. Palestinian midwives in Gaza were not only the first respondents but often the only trained professionals available to assist women to give birth safely in tents and shelters under intense, ongoing airstrikes and bombardments. The increasing violation of IHL during modern warfare leaves civilians and health care providers unprotected.

The study highlighted how midwives applied their competencies and critical thinking, adapting to normal physiological methods to avoid complications. It recommends the development of clear policies aligned with midwives' scope of practice that support their professional role when caring for pregnant women in war, emergency and humanitarian contexts. Given the increased instability globally, we believe the lessons learned from midwives in Gaza will be useful for any preparedness strategies. Midwives globally should be trained and equipped to assist women giving birth outside hospitals and be actively engaged in preparedness strategies for emergency response. Midwives need guidance, support, necessary supplies, connection to backup resources and training to assist births safely outside hospital settings. The IHL must be restored to protect civilians and healthcare providers during armed conflict.

## Supporting information

**S1 Appendix. Example of data extract containing units of data and line-by-line coding.**
(DOCX)

**S2 Appendix. Example of development from colliding codes to sub-themes and themes.**
(DOCX)

**S3 Appendix. Interview Guide (in English).**
(DOCX)

**S4 Appendix. Informed Consent Form (in English).**
(DOCX)

**S1 File. COREQ (COnsolidated criteria for REporting Qualitative research) Checklist.**
(PDF)

**S2 File. Inclusivity-in-global-research-questionnaire.**
(DOCX)

## Acknowledgments

The authors wish to thank all midwives for sharing their stories, experiences and insights.
**Patient consent**: The study did not involve patients. Not required

## Author contributions

**Conceptualization:** Sahar Hassan, Suha Baloushah, Berit Mortensen.

**Data curation:** Sahar Hassan, Suha Baloushah.

**Formal analysis:** Sahar Hassan, Suha Baloushah, Berit Mortensen.

**Funding acquisition:** Sahar Hassan, Berit Mortensen.

**Methodology:** Sahar Hassan, Suha Baloushah.

**Project administration:** Sahar Hassan.

**Resources:** Sahar Hassan.

**Software:** Sahar Hassan, Suha Baloushah, Berit Mortensen.

**Supervision:** Sahar Hassan.

**Validation:** Sahar Hassan, Suha Baloushah.

**Visualization:** Sahar Hassan.

**Writing – original draft:** Sahar Hassan, Berit Mortensen.

**Writing – review & editing:** Sahar Hassan, Suha Baloushah, Berit Mortensen.

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
