## [Decision Letter · Decision Letter 0]

6 Apr 2026

PONE-D-25-65361Birth in Shelters: Midwives’ Lived Experiences in providing childbirth care Amidst war in GazaPLOS One

Dear Dr. Hassan,

Thank you for submitting your manuscript to PLOS ONE. After careful consideration, we feel that it has merit but does not fully meet PLOS ONE’s publication criteria as it currently stands. Therefore, we invite you to submit a revised version of the manuscript that addresses the points raised during the review process.

Two reports have been obtained. Please address esteemed reviewers' comments in your revision.

We look forward to receiving your revised manuscript.

Kind regards,

Muhammad Haroon Stanikzai

Academic Editor

PLOS One

Journal Requirements:

2. In the ethics statement in the Methods, you have specified that verbal consent was obtained. Please provide additional details regarding how this consent was documented and witnessed, and state whether this was approved by the IRB

3. Please include a complete copy of PLOS’ questionnaire on inclusivity in global research in your revised manuscript. Our policy for research in this area aims to improve transparency in the reporting of research performed outside of researchers’ own country or community. The policy applies to researchers who have travelled to a different country to conduct research, research with Indigenous populations or their lands, and research on cultural artefacts. The questionnaire can also be requested at the journal’s discretion for any other submissions, even if these conditions are not met.  Please find more information on the policy and a link to download a blank copy of the questionnaire here: https://journals.plos.org/plosone/s/best-practices-in-research-reporting. Please upload a completed version of your questionnaire as Supporting Information when you resubmit your manuscript.

Reviewers' comments:

Reviewer's Responses to Questions

**Comments to the Author**

1. Is the manuscript technically sound, and do the data support the conclusions?

Reviewer #1: Yes

Reviewer #2: Yes

2. Has the statistical analysis been performed appropriately and rigorously? 

Reviewer #1: Yes

Reviewer #2: N/A

3. Have the authors made all data underlying the findings in their manuscript fully available?

Reviewer #1: No

Reviewer #2: Yes

4. Is the manuscript presented in an intelligible fashion and written in standard English?

Reviewer #1: Yes

Reviewer #2: Yes

5. Review Comments to the Author

Reviewer #1: The article is well written and methodologically clear. I found the themes could be better titled for example Theme 1: Unprotected in ruthless warfare could be Midwifes Safety or something similar. There is a known war that the authors refer to but there is often a tendency to add adjectives to explain the brutality of the war e.g. ruthless war etc. I would advise the authors to consider reframing the war without adjectives. Moreover, as qualitative researchers, if there are any prior experiences or biases that may influence the interpretation of the results, that needs to be mentioned. This is often encouraged in any qualitative research. The second theme has a very large results description which either needs to be shortened or another sub theme introduced. Do the midwives receive any form of prior mental health training to provide support given there is a history of conflict in the area being served? Lastly I wondered why the midwives didn't share more insights of birth complications example PPH or eclampsia or obstructed labour as a challenge in the circumstances. The results and discussion expand on lack of services which gives the reader a sense of the challenges but I wondered if the authors can comment on this query?

Reviewer #2: Thank you, editorial team, inviting me to review this manuscript. This is a methodologically sound and ethically important qualitative study. The authors present a compelling narrative that is both scientifically rigorous and deeply human. The findings have critical implications for humanitarian policy, midwifery education, and the protection of healthcare workers in conflict zones. I have several comments and questions aimed at strengthening the manuscript's clarity, methodological transparency, and the contextualization of its findings before it is ready for publication.

1. Abstract

1.1 The abstract states, "Within two years, most of the 2.1 million inhabitants were displaced and more than 69 500 people were directly killed." The timeline in the main text (page 2) clarifies the war started in October 2023 and the study was conducted from January to April 2025. Can you please specify the exact dates for the "two years" mentioned in the abstract to ensure consistency with the detailed timeline provided in the manuscript?

1.2 In the abstract, you mention "three additional themes." Please ensure the three themes listed in the abstract ("Unprotected...", "Professional role...", "Challenges and potentials...") exactly match the themes as they appear in the results section of the main text for consistency.

2. Introduction

2.1 On page 2, you cite that "more than 69 500 people were directly killed, whom more than 70% are women and children." The data source (Jamaluddine et al., 2025) is a capture-recapture analysis. To enhance the scientific robustness of this statistic in the introduction, could you briefly add a note on the methodology used to arrive at this figure (e.g., "using a capture-recapture analysis, which accounts for underreporting...")?

2.2 On page 4, you state that "IHL... are meant to protect civilians and health care providers during war." Given the extensive documentation in your paper of attacks on healthcare, including the targeting of hospitals and the displacement of midwives, do you think a brief, explicit statement about the perceived failure of IHL in this specific context would strengthen the argument for why this study is urgently needed?

3. Material and Methods

3.1 In page 5: You mention "no personally identifiable information was recorded." However, you also state interviews were conducted "face-to-face." How did you ensure that the location and time of the interview, as well as the participants' visible characteristics, did not inadvertently compromise their anonymity, especially given the small sample size and the close-knit, displaced community context?

3.2 In page 6: You used Perplexity AI to translate the transcripts for the third author. This is a novel approach. To ensure methodological transparency and address potential concerns about accuracy or bias, could you please elaborate on how the translated English transcripts were validated against the original Arabic? You mention they were "confirmed by the co-authors," but a more detailed step (e.g., "the two Arabic-speaking authors reviewed and approved the translated transcripts for accuracy and contextual meaning before the third author began analysis") would strengthen this section.

3.3 Table 1 (page 10): The table lists years of experience, but the narrative text mentions an average of \(11.0 \pm 5.93\) years. For Midwife 9, the table lists 28 years, which appears to be a significant outlier. Is this correct? If so, its impact on the analysis and findings should be noted. Also, Midwife 2 is listed as having 5 years of experience, but her narrative (MW2) is particularly rich and central to the overarching theme. It would be useful to briefly note in the text that participants spanned a wide range of experience, which contributed to the richness of the data.

4. Results

4.1 Theme 1, subtheme "Improvised Childbirth" (page 11): MW9's story of using a blanket from a vegetable cart and a donkey cart for transport is incredibly powerful. Did any midwife recount a situation where improvisation failed, leading to a poor outcome (e.g., maternal or neonatal death, infection)? While your findings focus on successful outcomes, acknowledging the presence or absence of such stories is important for a complete picture. You mention this as a limitation, but a note in the results about the nature of the narratives (e.g., "All narratives shared by the midwives concluded with a live birth, though complications were managed") would be helpful.

4.2 Theme 2, subtheme "Improvising preparedness" (page 13): The quote about using a kitchen knife and sewing thread is stark. You mention that after some incidents, there was a focus on ensuring medical points had delivery kits. Can you provide more detail? What was the nature of this "focus"? Was it a formal initiative by the Ministry of Health, UNRWA, or a grassroots effort among the midwives themselves? This small detail adds significant depth to the theme of "challenges and potentials."

4.3 Theme 3, subtheme "Navigating high-risk pregnancies" (page 16): The quote from MW8 about the woman with a previous C-section and a "precious baby" after infertility highlights the immense pressure on midwives. Was there any instance where a midwife had to make a critical decision, such as deciding not to attempt a home birth for a clearly high-risk patient, and if so, how did they manage that situation without access to a hospital? Exploring the decision-making process in these extreme scenarios would add further depth.

5. Discussion

5.1 Page 18: You state that "midwives during the recent war did not mention fear of restrictions, which might imply a growing recognition of the profession's autonomy." This is a crucial point. Could it also be that the complete breakdown of the formal health system and the state of emergency simply rendered legal restrictions moot? The authors might explore this nuance further: was the absence of fear due to autonomy or anarchy? This distinction has significant implications for how preparedness policies are framed (i.e., empowering vs. emergency protocols).

5.2 Page 18-19: You reference the Quality Maternal and Newborn Care (QMNC) framework to highlight how midwives' care aligned with its philosophy. The most urgent missing quality was "sufficient for first line management of complications." This is a critical gap. In your discussion, could you elaborate on what specific "first-line" supplies (e.g., misoprostol for PPH, antibiotics, magnesium sulfate) were most urgently needed and how the absence of these impacted the midwives' ability to manage the high-risk cases you described? This would make the recommendation for "necessary supplies" in the conclusion more specific and actionable.

5.3 Page 20: You mention the UNRWA's role and its being targeted. This is a critical part of the context. Could you briefly expand on how the targeting of UNRWA and the subsequent disruption of its supply chains directly affected the midwives you interviewed? Did they mention a specific point at which they lost access to UNRWA supplies? This would strengthen the link between geopolitical events and the on-the-ground experiences of your participants.

6. Strengths and Limitations

6.1 Page 20: The stated limitation is that "all participating midwives shared stories that, in some way, ended well with a healthy mother and baby. Important information may have refrained from participating due to fear of being perceived as having failed." This is an important and honest limitation. Could the authors expand on this by discussing the potential impact of this selection bias on the findings? For instance, does this mean the findings represent a "best-case scenario" of midwifery response, and that the experiences of those who faced tragic outcomes might reveal different, perhaps even more critical, failures in the system? Acknowledging this gap explicitly would be valuable.

7. Conclusion and Recommendations

7.1 Based on your findings, what would be the first, most critical element of such a policy? Is it the provision of a standardized "emergency birth kit" with essential supplies? Is it a legal framework that explicitly authorizes and protects midwives practicing in community settings during declared emergencies? Prioritizing this in the final sentence of your conclusion could give your recommendation more immediate impact.

6. PLOS authors have the option to publish the peer review history of their article (what does this mean?). If published, this will include your full peer review and any attached files.

Reviewer #1: **Yes:** Dr Danish Ahmad MBBS, MSc PhD,MNAMS, FRCP(Edin) FRCP(Lon) FHEA (UK)

Reviewer #2: **Yes:** Temesgen Anjulo Ageru

You may also use PLOS’s free figure tool, NAAS, to help you prepare publication quality figures: https://journals.plos.org/plosone/s/figures#loc-tools-for-figure-preparation

---

## [Author Response · Author response to Decision Letter 1]

22 Apr 2026

Point-by-point response to reviewers’ comments

Reviewers' comments:

Reviewer #1: The article is well written and methodologically clear.

I found the themes could be better titled for example Theme 1: Unprotected in ruthless warfare could be Midwifes Safety or something similar.

Response: Thank you for this valuable suggestion. We appreciate the recommendation to simplify theme titles for clarity. While we agree that titles such as “Midwives’ Safety” are more concise, we intentionally retained “Unprotected in ruthless warfare” to reflect the intensity and lived experiences expressed by midwives.

The analytic process involved extensive and iterative discussions among the three authors, culminating in consensus on theme names that are both conceptually grounded and contextually rich. The authors were deeply immersed in the data over several months, engaging in repeated reading, re-reading, and collective reflection on midwives’ narratives.

We aimed for the theme titles to capture not only issues of safety, but also what midwives experienced and expressed in their own terms. The notion of being “unprotected” while providing care to pregnant women emerged strongly from the data and conveys the reality participants sought to communicate. In this context, midwives described working in an environment where safety could not be assured, even while delivering essential care. For these reasons, we respectfully prefer to retain the original theme titles, as they were derived through a rigorous and collaborative analytic process and are intended to faithfully represent participants’ voices and the depth of the context.

There is a known war that the authors refer to but there is often a tendency to add adjectives to explain the brutality of the war e.g. ruthless war etc. I would advise the authors to consider reframing the war without adjectives.

Response: Thank you for this comment. We acknowledge that this war has been experienced by midwives as exceptionally severe and devastating, with widespread impact on civilians, including women and children. One of the authors (the second author) has lived through these conditions daily, which inevitably shaped the research context and the team’s engagement with the data.

We recognize the importance of maintaining academic rigor and minimizing bias. At the same time, our use of descriptive language was grounded in participants’ narratives and intended to faithfully reflect their lived experiences. We have therefore aimed to balance neutrality with an accurate representation of the intensity and context conveyed by these midwives.

However, we have revised the term “war” throughout the manuscript and removed unnecessary adjectives (lines 83, 298, 575, 594, 597, 635, 646, and 674). However, we retained the adjective in the first theme, as we consider it essential to faithfully reflect midwives’ voices and the contextual depth. We hope this satisfies the reviewer’s inquiry. Modified.

Moreover, as qualitative researchers, if there are any prior experiences or biases that may influence the interpretation of the results, that needs to be mentioned. This is often encouraged in any qualitative research.

Response: Thank you for the comment. We have reflected on the author's previous experiences, and elaborated on credibility and dependability (page 6, lines 192-196) and (page 8, lines 234-236). No modifications were needed.

The second theme has a very large results description, which either needs to be shortened or another sub-theme introduced.

Response: Thank you for the suggestion. The second theme included five subthemes related to the professional role and associated emotional tensions, which were presented sequentially and supported with appropriate quotations. These subthemes were: midwifery practice and confidence, improvised preparedness, accessibility and availability among the displaced, balancing personal and professional life during war, and midwives’ emotional strength and faith.

Our initial analysis generated 6–8 themes. However, through iterative revisions, in-depth discussions, and extensive conceptual work among the three authors, we reached a consensus to consolidate these into five themes and ultimately into three overarching themes. Arriving at these three themes required substantial analytical effort to ensure they were comprehensive while avoiding redundancy.

The presentation of Theme 2 follows these five subthemes, with italics used at the beginning of each paragraph to signal a new subtheme while maintaining a coherent narrative flow. With respect to the reviewer’s suggestion, we prefer to retain the current structure to avoid redundancy and unnecessary repetition across themes. No modifications were needed.

Do the midwives receive any form of prior mental health training to provide support given there is a history of conflict in the area being served?

Response: Thank you for this question. In general, training for healthcare providers is largely dependent on the availability of external funding, and there is no institutionalized in-service training in either the West Bank or Gaza. In Gaza, midwives working in primary healthcare clinics have received some training in mental health; however, midwives working in hospitals do not receive such training, as available funding is primarily directed toward primary healthcare settings. No modifications were needed.

Lastly, I wondered why the midwives didn't share more insights of birth complications example PPH or eclampsia or obstructed labour as a challenge in the circumstances.

Response: Thank you for this important note. During the research team discussions, we also reflected on why midwives did not share situations involving complications. We believe that midwives may have been reluctant to disclose such cases, as assisting women to deliver outside the hospital is considered an illegal practice, and midwives are not permitted to provide this type of care even in a crisis situation, as there is no policy in place. Consequently, they may have feared potential repercussions or sanctions from the health system. It is therefore possible that midwives selectively shared narratives that ended without complications.

Notably, two midwives reported assisting a labouring woman with history of infertility and a previous cesarean section, as the war was intense and both cases had no access to an ambulance to be transferred to the hospital. We did include relevant quotations from midwives (MW7 and MW8; page 18, lines 546–551, and page 19, lines 556-558), who reported assisting women with potential risks. Similarly, MW8 reported assisting a woman with a prior history of postpartum hemorrhage at a focal medical point in a school, where she was the most qualified provider and ambulance access was not possible. In this situation, Pitocin was administered intravenously, as it was available at the site. We also acknowledged this in the limitations section (page 21, lines 659-662) and on (page 21, line 662-666), we added the following:

“Additionally, midwives may have been reluctant to disclose cases with complications, as assisting women to deliver outside the hospital is considered an illegal practice. Consequently, they may have feared potential consequences or sanctions from the health system. It is therefore possible that midwives selectively shared narratives that ended without complications.”

We hope this elaboration and modification satisfy the reviewer’s comment. Modified.

The results and discussion expand on lack of services which gives the reader a sense of the challenges but I wondered if the authors can comment on this query?

Response: Thank you for this query. Displaced populations were indeed left without adequate services, as the health system is primarily structured around hospitals and primary healthcare clinics provided by the two main sectors: the Ministry of Health and UNRWA. During periods of intense airstrikes and evacuation orders, UNRWA clinics, despite being widely distributed, often shut down. Access to hospitals was also severely constrained due to a lack of transportation, destruction of roads, and ongoing hostilities.

Within displaced communities, some grassroots groups and local NGOs established focal medical points, typically run by members of the affected communities. Many of these points operated with few or no physicians, nurses, or midwives, and were equipped only with basic supplies such as cotton, gauze, antiseptics, common analgesics, and occasionally antibiotics.

Midwives were still required to attend their scheduled hospital shifts despite the risks. Some reported walking for hours under active airstrikes to reach their workplaces, as they had no alternative if they wished to retain their positions and secure a portion of their salary to support their families. Outside their official working hours, particularly in the evenings and at night, midwives were often approached by pregnant women within their communities for assistance.

A small number of midwives reported receiving one or two disposable delivery kits, assembled and distributed by local NGOs; however, this was insufficient, and many midwives did not receive any kits at all. We hope this elaboration clarifies the reviewer’s query. No modifications were needed.

Reviewer #2: Thank you, editorial team, inviting me to review this manuscript. This is a methodologically sound and ethically important qualitative study. The authors present a compelling narrative that is both scientifically rigorous and deeply human. The findings have critical implications for humanitarian policy, midwifery education, and the protection of healthcare workers in conflict zones. I have several comments and questions aimed at strengthening the manuscript's clarity, methodological transparency, and the contextualization of its findings before it is ready for publication.

Response: Thank you for your time and effort.

1. Abstract

1.1 The abstract states, "Within two years, most of the 2.1 million inhabitants were displaced and more than 69 500 people were directly killed." The timeline in the main text (page 2) clarifies the war started in October 2023 and the study was conducted from January to April 2025. Can you please specify the exact dates for the "two years" mentioned in the abstract to ensure consistency with the detailed timeline provided in the manuscript?

Response: Thank you for the note. The two years we refer to are from October 2023 until the time of writing this manuscript (October 2025). We have clarified this in the abstract. (Page 2, line 40). Modified.

1.2 In the abstract, you mention "three additional themes." Please ensure the three themes listed in the abstract ("Unprotected...", "Professional role...", "Challenges and potentials...") exactly match the themes as they appear in the results section of the main text for consistency.

Response: We have revised the themes throughout the manuscript. They appear the same in the abstract, in the results section, and in Figure 1. No modifications were needed.

2. Introduction

2.1 On page 2, you cite that "more than 69 500 people were directly killed, whom more than 70% are women and children." The data source (Jamaluddine et al., 2025) is a capture-recapture analysis. To enhance the scientific robustness of this statistic in the introduction, could you briefly add a note on the methodology used to arrive at this figure (e.g., "using a capture-recapture analysis, which accounts for underreporting...")?

Response: Thank you for this query. We are referring to the figure reported in the OCHA report for November 2025 (69,513) (Reference below and cited). OCHA cites this number as reported by the Ministry of Health (MoH), which can be considered a trusted source. However, this figure is likely an underestimate, as many individuals remain unaccounted for under the rubble.

Reference: OCHA. Humanitarian Situation Update #342 | Gaza Strip: OCHA; Nov 2025 [cited 2025 7/12/2025]. Available from: https://www.ochaopt.org/content/humanitarian-situation-update-342-gaza-strip. No modifications were needed.

2.2 On page 4, you state that "IHL... are meant to protect civilians and health care providers during war." Given the extensive documentation in your paper of attacks on healthcare, including the targeting of hospitals and the displacement of midwives, do you think a brief, explicit statement about the perceived failure of IHL in this specific context would strengthen the argument for why this study is urgently needed?

Response: Thank you for the comment. We added this sentence:

“This study is crucial as it generates context-specific evidence on delivering childbirth care in extreme and resource-constrained conflict settings, where formal health systems are disrupted”. (Page 5, lines 149-151). We hope this is satisfactory. Modified.

3. Material and Methods

3.1 In page 5: You mention "no personally identifiable information was recorded." However, you also state interviews were conducted "face-to-face." How did you ensure that the location and time of the interview, as well as the participants' visible characteristics, did not inadvertently compromise their anonymity, especially given the small sample size and the close-knit, displaced community context?

Response: Thank you for this note. It is correct that some participants were identified by the interviewer (second author), who is a midwife residing in Gaza, and that the interviews were conducted face-to-face. However, no personal identifying information was collected or recorded during the interviews. For example, no names, phone numbers, identification numbers, or places of residence were mentioned in the recordings.

This ensured that the data remained anonymous to the other two authors. Additionally, many participants were not previously known to the interviewer prior to recruitment. We hope this clarification helps to contextualize the process and address the reviewer’s concern. No modifications were needed.

3.2 In page 6: You used Perplexity AI to translate the transcripts for the third author. This is a novel approach. To ensure methodological transparency and address potential concerns about accuracy or bias, could you please elaborate on how the translated English transcripts were validated against the original Arabic? You mention they were "confirmed by the co-authors," but a more detailed step (e.g., "the two Arabic-speaking authors reviewed and approved the translated transcripts for accuracy and contextual meaning before the third author began analysis") would strengthen this section.

Response: Thank you for the suggestion.

The sentence “The two Arabic-speaking authors reviewed and approved the translated transcripts for accuracy and contextual meaning before the third author initiated the analysis.” Was added. (Page 7, line 225-227). Modified.

3.3 Table 1 (page 10): The table lists years of experience, but the narrative text mentions an average of \(11.0 \pm 5.93\) years. For Midwife 9, the table lists 28 years, which appears to be a significant outlier. Is this correct? If so, its impact on the analysis and findings should be noted. Also, Midwife 2 is listed as having 5 years of experience, but her narrative (MW2) is particularly rich and central to the overarching theme. It would be useful to briefly note in the text that participants spanned a wide range of experience, which contributed to the richness of the data.

Response: Thank you for the important comment. Upon our review of the original data, the participant reported 28 years of experience; however, she also stated that she graduated in 1999 and has been working since then, which corresponds to 26 years of experience as a midwife. This has been revised and corrected in both the table and the text.

A sentence was modified as follows:

“The midwives’ experience varied widely (mean ± SD: 12.67 ± 7.48 years), contributing to the richness of the data.” (Page 9, lines 272-273). Modified.

4. Results

4.1 Theme 1, subtheme "Improvised Childbirth" (page 11): MW9's story of using a blanket from a vegetable cart and a donkey cart for transport is incredibly powerful. Did any midwife re

---

## [Decision Letter · Decision Letter 1]

8 May 2026

Birth in Shelters: Midwives’ lived experiences in providing childbirth care amidst war in Gaza

PONE-D-25-65361R1

Dear Dr. Hassan,

We’re pleased to inform you that your manuscript has been judged scientifically suitable for publication and will be formally accepted for publication once it meets all outstanding technical requirements.

Kind regards,

Muhammad Haroon Stanikzai

Academic Editor

PLOS One

Additional Editor Comments (optional):

This manuscript presents a timely and methodologically sound work and I congratulate the authors for their work.

Reviewers' comments:

Reviewer's Responses to Questions

**Comments to the Author**

1. If the authors have adequately addressed your comments raised in a previous round of review and you feel that this manuscript is now acceptable for publication, you may indicate that here to bypass the “Comments to the Author” section, enter your conflict of interest statement in the “Confidential to Editor” section, and submit your "Accept" recommendation.

Reviewer #1: All comments have been addressed

Reviewer #2: All comments have been addressed

2. Is the manuscript technically sound, and do the data support the conclusions?

Reviewer #1: Yes

Reviewer #2: Yes

3. Has the statistical analysis been performed appropriately and rigorously? 

Reviewer #1: N/A

Reviewer #2: Yes

4. Have the authors made all data underlying the findings in their manuscript fully available?

Reviewer #1: No

Reviewer #2: Yes

5. Is the manuscript presented in an intelligible fashion and written in standard English?

Reviewer #1: Yes

Reviewer #2: Yes

6. Review Comments to the Author

Reviewer #1: The authors have addressed my comments and also the 2nd reviewer points. I am largely happy to recommend the paper for publication given it is a qualitative paper with established methodology and the availability of timely results would be important from the journals perspective.

Reviewer #2: The authors' responses are respectful, evidence-based, and have genuinely improved the manuscript's rigor, transparency, and contextual depth. No outstanding issues remain from my previous review.

7. PLOS authors have the option to publish the peer review history of their article (what does this mean?). If published, this will include your full peer review and any attached files.

Reviewer #1: **Yes:** Dr Danish Ahmad, MBBS,MSc,PhD, FRCP

Reviewer #2: **Yes:** Temesgen Anjulo Ageru

---

## [Editor Report · Acceptance letter]

PONE-D-25-65361R1

PLOS One

Dear Dr. Hassan,

I'm pleased to inform you that your manuscript has been deemed suitable for publication in PLOS One. Congratulations! Your manuscript is now being handed over to our production team.

Kind regards,

on behalf of

Dr. Muhammad Haroon Stanikzai

Academic Editor

PLOS One